# A Simple Frailty Score Predicts Survival and Early Mortality in Systemic AL Amyloidosis

**DOI:** 10.3390/cancers16091689

**Published:** 2024-04-26

**Authors:** Rafael Ríos-Tamayo, Ramón Lecumberri, María Teresa Cibeira, Verónica González-Calle, Rafael Alonso, Amalia Domingo-González, Elena Landete, Cristina Encinas, Belén Iñigo, María-Jesús Blanchard, Elena Alejo, Isabel Krsnik, Manuel Gómez-Bueno, Pablo Garcia-Pavia, Javier Segovia-Cubero, Laura Rosiñol, Juan-José Lahuerta, Joaquín Martínez-López, Joan Bladé

**Affiliations:** 1Hospital Universitario Puerta de Hierro Majadahonda, IDIPHISA, CIBERCV, 28222 Madrid, Spain; 2Clínica Universidad de Navarra, CCUN, IDISNA, Universidad de Navarra, 31008 Pamplona, Spain; 3Hospital Clinic de Barcelona, IDIBAPS, Universitat de Barcelona, 08007 Barcelona, Spain; 4University Hospital of Salamanca (HUS/IBSAL), CIBERONC, Center for Cancer Research-IBMCC (USAL-CSIC), 37007 Salamanca, Spain; 5Hospital Universitario 12 de Octubre, Instituto de Investigación del Hospital Universitario 12 de Octubre, 28041 Madrid, Spain; 6Hospital Universitario Fundación Jiménez Díaz, 28040 Madrid, Spain; 7Hospital Universitario Infanta Leonor, 28040 Madrid, Spain; 8Hospital General Universitario Gregorio Marañón, IiSGM, 28009 Madrid, Spain; 9Hospital Clínico San Carlos, 28040 Madrid, Spain; 10Hospital Universitario Ramón y Cajal, 28034 Madrid, Spain; 11Centro Nacional de Investigaciones Cardiovasculares (CNIC), 28029 Madrid, Spain

**Keywords:** AL amyloidosis, survival, early mortality, frailty

## Abstract

**Simple Summary:**

Despite a lack of standardization and some open questions, a growing body of evidence supports the use of frailty to optimize the clinical management of patients with hematological malignancies. Several scores have been applied particularly to multiple myeloma, an entity that shares many characteristics with AL amyloidosis, both being frequently associated. To date, no study has focused on frailty in patients with AL amyloidosis. We aimed to define a practical evaluation of frailty and estimate its impact in survival in patients with systemic AL amyloidosis.

**Abstract:**

Systemic AL amyloidosis is a challenging disease for which many patients are considered frail in daily clinical practice. However, no study has so far addressed frailty and its impact on the outcome of these patients. We built a simple score to predict mortality based on three frailty-associated variables: age, ECOG performance status (<2 vs. ≥2) and NT-proBNP (<8500 vs. ≥8500 ng/L). Four-hundred and sixteen consecutive newly diagnosed patients diagnosed at ten sites from the Spanish Myeloma Group were eligible for the study. The score was developed in a derivation cohort from a referral center, and it was externally validated in a multicenter cohort. Multivariate analysis showed that the three variables were independent predictors of survival. The score was able to discriminate four groups of patients in terms of overall survival and early mortality in both cohorts. Comorbidity was also analyzed with the Charlson comorbidity index, but it did not reach statistical significance in the model. A nomogram was created to easily estimate the mortality risk of each patient at each time point. This score is a simple, robust, and efficient approach to dynamically assess frailty-dependent mortality both at diagnosis and throughout follow-up. The optimal treatment for frail AL amyloidosis patients remains to be determined but we suggest that the estimation of frailty-associated risk could complement current staging systems, adding value in clinical decision-making in this complex scenario.

## 1. Introduction

Systemic light chain (AL) amyloidosis is a protein misfolding disease caused by a plasma-cell clone that produces toxic light chains with the ability to aggregate and deposit in target organs forming amyloid and leading to progressive organ dysfunction [1,2,3]. The incidence of AL amyloidosis is increasing over time in some recent population-based studies [4,5]. The prevalence is getting higher due to improved therapies and their positive impact on overall survival (OS) [4,6,7]. Despite this, the gap in OS compared with the demographically matched general population is still remarkable [4]. However, outcomes of patients are very variable depending on different prognostic factors either related to the patient or the disease. Among disease-related prognostic factors, we can distinguish between those associated with the tumor burden, such as serum free light chain (FLC) concentration and bone marrow plasma cell infiltration, those associated with the biological characteristics of the tumor clone, such as cytogenetic abnormalities, and the pattern of organ involvement, particularly of the heart (very well displayed by cardiac biomarkers) [8,9,10]. Regarding patient-related factors, they classically include age, performance status and comorbidity, although other psycho-social aspects can also influence prognosis [11,12,13]. Among all these prognostic factors, the currently used staging systems initially included troponin and N-terminal fragment of the pro-brain natriuretic peptide (NT-proBNP), and later added the serum FLC difference, as a measure of severity of cardiac involvement and tumor burden, respectively [14,15,16]. Another key prognostic factor is response to therapy [17,18,19,20], which is unavailable at the time of diagnosis. Overall, the prognostic evaluation of AL amyloidosis patients is a rapidly changing field. Therefore, a comprehensive prognostic assessment is an increasingly complex task in the real-world setting. 

Cardiac amyloidosis plays a critical role in the global management of systemic AL amyloidosis [21], provided that the heart is the most frequently involved organ and the extent of its involvement carries a tremendous prognostic impact, even in patients with concurrent multiple myeloma (MM) [22]. Patients with advanced cardiac stages have poor OS and a high rate of early mortality. Strikingly, elderly patients with poor performance status and advanced cardiac disease are commonly considered extremely frail, raising the need to tailor therapy to patient’s frailty [3].

Frailty assessment should help to individualize and optimize therapy, build risk-based realistic clinical goals, avoid under-treating fit or over-treating frail patients (with potential unacceptable toxicity), adapt supportive treatment, and adjust the multidisciplinary approach. There is a growing body of evidence about the need to measure frailty in patients with hematologic malignancies and there are expanding data in particular entities such as MM [23,24,25,26,27,28,29,30,31,32,33]. In this sense, Milani et al. [28] developed a simple frailty score based on age ≥ 70 years, Eastern Cooperative Oncology Group (ECOG) performance status ≥ 2, and NT-proBNP ≥ 300 ng/L, that was able to discriminate four groups of patients with different OS in a series of patients with MM from the Mayo Clinic. So far there are no data available about frailty in AL amyloidosis.

With this background, we sought to develop a simple and easy-to-use score to estimate frailty-based mortality in systemic AL amyloidosis, with the aim to contribute to the development of evidence-based frailty-adapted treatment strategies for this disease.

## 2. Materials and Methods

### 2.1. Study Design

A retrospective observational study was performed using prospectively collected data from patients with systemic AL amyloidosis that were evaluated at ten centers belonging to the GEM-PETHEMA (Spanish Myeloma Group). Based on epidemiological data, staging systems, and a frailty score developed for MM, we built a new frailty score for AL amyloidosis. A nomogram was created to estimate the probability of frailty-derived early death and OS for each patient. The score was developed in a derivation cohort of a referral center. Afterwards, it was externally validated in an independent multicenter cohort of 9 medical centers. 

The analysis was performed separately in two periods of time, 2005–2014 and 2015–2023, to assess potential changes in OS over time.

### 2.2. Patients

The derivation cohort consisted of all patients with systemic AL amyloidosis consecutively diagnosed and treated at a referral center and enrolled in a prospectively maintained registry from January 2005 to August 2023. The validation cohort included patients with systemic AL amyloidosis diagnosed during the same period of time at nine centers of the Spanish Myeloma Group. Additional eligibility criteria for both cohorts in this study were: biopsy-proved systemic AL amyloidosis, baseline availability of ECOG and NT-proBNP, and a minimum follow-up of 6 months. All patients had consented the use of their medical records. The study was conducted in accordance with the institutional guidelines with approval of the institutional review board and in accordance with the principles of the Declaration of Helsinki. All data accessed complied with current data protection and privacy regulations.

### 2.3. End-Points

The main end-point of the study is OS, which was measured from the time of diagnosis to death for any cause or last date known to be alive for censored patients. Early mortality was evaluated at 3, 6, and 12 months.

### 2.4. Variables

Both clinical and laboratory variables of prognostic interest were prospectively collected. 

Diagnostic delay was calculated from the date of the first related symptom to the date of the diagnostic biopsy. Therapeutic delay was determined from the date of the diagnostic biopsy to the first day of treatment. 

Comorbidities were studied by two methods. First, a comprehensive approach as previously described [34], including all potentially relevant baseline comorbidities, analyzing both the number of comorbidities as well as the individualized impact of each comorbidity. In addition, the Charlson comorbidity index (CCI) was also analyzed. 

Organ involvement was assessed according to standard criteria [17].

### 2.5. Statistics

Categorical variables were reported as frequencies and percentages by category, and continuous variables were summarized using descriptive statistics. Comparisons for categorical variables among different groups were made with the χ^2^-test, using Fisher’s exact test when appropriate. Comparisons of means of quantitative continuous variables between the two groups were made with the *t*-test or Mann–Whitney U test. Median follow-up was estimated according to the reverse Kaplan–Meier method. Median OS with a 95% confidence interval (CI) was estimated using the Kaplan–Meier method, and comparisons among groups were carried out with the log-rank test. Cox proportional hazards were used for the calculation of hazard ratios for each variable. For multivariable analysis, factors with statistical significance at the 0.1 level were introduced into a Cox proportional hazards model (backward analysis). Discrimination was assessed through the c-index and calibration by means of a calibration plot. All *p*-values were two-sided. No imputation for missing data was used. Data were analyzed with Stata v18 and SPSS v20 software. The threshold for statistical significance was set at *p* value ≤ 0.05.

## 3. Results

### 3.1. Derivation Cohort

The derivation cohort consisted of 134 patients after excluding 15 who did not meet the eligibility criteria of the study (Figure 1). The cohort included 71 males and 63 females (53% vs. 47%), and the median age was 64.5 years (IQR 55–72.3). The baseline characteristics of the series are shown in Table 1. 

Remarkably, the heart was involved in most patients (91%), with NT-proBNP ≥ 8500 ng/L in 40.3%, whereas kidney was the second most frequently involved organ (47%). The high rate of severe cardiac involvement can be explained by the features of our institution which is a hematologic and cardiac referral center. Median diagnostic delay was 6 months (IQR 4–12) and median therapeutic delay 18 days (IQR 8–32.5). First line of therapy was given to 126 patients and included bortezomib in 73 (55.7%), daratumumab in 23 (17.6%), and lenalidomide in 27 (20.6%). Thirty-three patients (24.6%) underwent high-dose melphalan and autologous stem cell transplant, 29 of them (87.9%) conditioned with melphalan 200 mg/m^2^.

Eight patients (6%) developed different secondary tumors (bladder, lung, skin, head and neck, breast, secondary myelodysplastic syndrome, secondary acute myeloid leukemia, unknown). 

At the time of this analysis, 63 patients remained alive (47%) and 71 had died. The main cause of death was progression or disease-related cause in 54/71 (76.1%) patients. 

Median follow-up was 62 months (95% CI, 50.8–73.2) whereas median OS was 52.6 months (95% CI, 24.8–80.4), 33.7 months (95% CI, 16.5–50.9) in the 2005–2014 period, and 67.8 months (95% CI, 45.9–89.7) in 2015–2023, *p* = 0.160.

### 3.2. Validation Cohort

The validation cohort was selected from a series of 462 patients diagnosed at nine Spanish centers. A total of 180 patients were not included in the study due to absence of baseline NT-proBNP available (138 individuals), unavailability of baseline ECOG (12 subjects), and diagnosis outside the time frame or follow-up of <6 months (30 subjects) (Figure 1). Thus, the final validation cohort comprised 282 patients, 149 (52.8%) of whom were male, and with a median age of 65 years (IQR 56–73). At the time of this analysis, 176 patients remained alive (62.4%) and 106 had died. Median follow-up of the series was 44.6 months (95% CI, 35.7–53.6) and median OS was 90.7 months (95% CI, 58.1–123.3), being 76.9 months (95% CI, 46.5–107.3) in the 2005–2014 period and not reached in the 2015–2023 period, *p* = 0.026. Globally, the median OS in the validation cohort was significantly longer (*p* = 0.012) than that estimated in the derivation cohort (Appendix A).

### 3.3. Comparison of Derivation and Validation Cohorts

The comparative characteristics of the three variables of the score in both cohorts are shown in Table 2. Age was similar in both cohorts, while there were striking differences in the other two variables. Regarding ECOG, 79.9% patients in the derivation cohort had an ECOG ≥ 2 compared to 30.1% in the validation cohort (*p* < 0.001). On the other hand, mean NT-proBNP was significantly higher in the derivation cohort (8919 vs. 4987.6 ng/L, *p* < 0.001), as it was the proportion of patients with NT-proBNP ≥ 8500 (40.3% vs. 17.7%, *p* < 0.001) (Appendix A).

### 3.4. The Frailty Score

The three variables of the model (with age categorized as <70 or ≥70 years for this purpose) had a very significant impact in terms of OS in both cohorts (Figure 2).

OS was estimated in both cohorts according to the three variables of the model (Figure 3). Four groups of patients are displayed with excellent discrimination in both cohorts (*p* < 0.001). Median OS was not reached in group I (no risk factor) for both cohorts. Patients in group II (one risk factor) presented median OS of 85.5 months (95% CI, 36.7–134.3) and 80.1 months (95% CI, 64.6–95.7), while group III (two risk factors) and group IV (all three risk factors) showed median OS of 14.4 months (95% CI, 0–40.3) and 3.8 months (95% CI, 2.1–5.5), and finally, 2.7 months (95% CI, 0–5.6) and 2.5 months (95% CI, 0–5.5) for the derivation and validation cohorts, respectively. Patients included in groups III and IV (55.2% in the derivation cohort and 22.3% in the validation cohort) presented a dismal outcome and could be considered as frailty-dependent high risk with poor OS and very high early mortality.

The univariate analysis and multivariable Cox regression model for the core frailty score are presented in Table 3. After applying the backward modeling strategy, the variables that remained in the final model were age, ECOG and NT-proBNP. Discrimination was good, with a Harrell’s C-index equal to 0.7370. A nomogram with the three independent variables of the model is shown in Figure 4. Both early mortality risk at 6 months as well as 24-month and 60-month OS can be easily and dynamically estimated for each newly diagnosed patient or at any time point during the follow-up, provided that all three variables can change over time. Calibration was evaluated at 6 and 60 months with acceptable agreement between observed and predicted event probabilities (Appendix A).

Regarding long-term survival, 105 of 416 patients (25.2%) in both cohorts lived ≥5 years, 34 of 134 (25.4%) in the derivation and 71 of 282 (25.2%) in the validation cohort, while 23 patients lived ≥10 years, 8 of 134 (6%) and 15 of 282 (5.3%) in the derivation and validation cohorts, respectively. In the group of 5-year survivors, 7 of 34 patients (20.6%) in the derivation cohort had both ECOG ≥ 2 and NT-proBNP ≥ 8500 ng/L whereas only 1 of 71 patients (1.4%) in the validation cohort had these characteristics (all of them younger than 70 years). In the group of 10-year survivors, only one patient in each group was found.

### 3.5. Ultra-Frail Patients

Independently of age, patients with both ECOG ≥ 2 and NT-proBNP ≥ 8500 ng/L represent the group with the poorest outcome. Eighty-nine out of 416 patients (21.4%) fulfilled these characteristics, and the median OS was 4 months (95% CI, 0.6–7.4). The distribution by cohorts was 52/134 (38.8%) in the derivation cohort and 37/282 (13.1%) in the validation cohort (*p* < 0.001), with median OS of 6.1 months (95% CI, 0–16.1) and 2.5 months (95% CI, 0.4–4.6), respectively (*p* = 0.141) (Appendix A). These patients could be considered a group of ultra-frail patients in terms of early mortality. However, even in this group with dismal prognosis, about 20% of patients can become long-term survivors.

### 3.6. The Role of Comorbidity

Comorbidity was only analyzed in the derivation cohort since its assessment is not standardized and it was only available in one of the nine centers of the validation cohort. Whole comorbidity data were missing in 4 patients (3%) of the derivation cohort. Regarding the CCI, 76 of 130 patients (58.5%) obtained a score of ≥3, with median OS of 33.7 (95% CI, 5.8–61.6) vs. 122.9 months (95% CI, 60.6–185.2) for those with <3 (*p* = 0.008). When CCI ≥3 was included in the core model of frailty, Harrell’s C-index increased to 0.7464, but it did not reach statistical significance (*p* = 0.253). However, this does not allow us to conclude that comorbidity plays no role in a frailty-based mortality risk score in systemic AL amyloidosis, because the CCI-based assessment of comorbidity is not considered the optimal approach since several key comorbidities are lacking. 

An optimized alternative to CCI including all potentially relevant comorbidities could offer a more realistic view (Appendix A). With this comprehensive strategy, 66 of 130 patients (50.8%) in the derivation cohort had ≥3 comorbidities, with a median OS of 34 (95% CI, 0–69.8) vs. 73.9 months (95% CI, 63.4–84.4) for those with <3 (*p* = 0.055). Moreover, 21 of 130 patients (16.2%) exhibited ≥5 comorbidities, with median OS of 11 (95% CI, 0.4–21.6) vs. 70.3 months (95% CI, 49.2–91.4) for those with <5 (*p* = 0.001). Besides the core score (age, ECOG ≥ 2, and NT-proBNP ≥ 8500 ng/L), the only variable that remains statistically significant in the multivariable Cox model is comorbidity, when using this comprehensive approach for patients with ≥5 comorbidities (Appendix A). Remarkably, several single comorbidities included in the comprehensive approach, such as arterial hypertension, hypocholesterolemia, cardiopathy (other than amyloidosis-derived cardiac involvement) or venous thromboembolism have a significant impact on OS, but they are not included in the CCI. The only single comorbidity which retained independent prognostic significance in the core score was venous thromboembolism. The inclusion of patients with a high comorbidity burden (≥5 comorbidities) in the final Cox model allows for better discrimination of the OS curves, but it remains to be standardized in daily clinical practice.

### 3.7. Early Mortality

Despite the contrast in baseline characteristics of patients, no statistically significant differences in terms of early mortality could be demonstrated between both cohorts over time. The early mortality rate at 3 months was 17.2% in the derivation cohort and 13.1% in the validation cohort (*p* = 0.297), while the rates at 6 months were 29.1% and 22.7% (*p* = 0.181), and at 12 months were 38.1% and 34.4%, respectively (*p* = 0.511).

When the analysis was performed separately in the two periods of time (2005–2014 and 2015–2023), early mortality showed a slight trend to decrease over time in both cohorts, without statistical significance. Thus, in the derivation cohort, mortality rates in the two periods were 19%/16.3% at 3 months, 31%/28.3% at 6 months, and 35.7%/39.1% at 12 months. In the validation cohort, those rates were 13.3%/10.9% at 3 months, 30%/22.7% at 6 months, and 30%/38.7% at 12 months (Appendix A).

The percentage of patients showing age ≥ 70 years, ECOG ≥ 2, and NT-proBNP ≥ 8500 ng/L along the three time points to assess early mortality (at 3, 6, and 12 months) in both cohorts, besides the percentage of patients with ≥5 comorbidities (in the derivation cohort), is presented in Appendix A, in order to estimate the relative impact of each variable during this critical period of time.

## 4. Discussion

A growing body of evidence supports considering frailty in the assessment and management of hematological malignancies. The missing link is for frailty scoring to translate into a simple and pragmatic tool for real-world clinical practice [32].

Despite there being no standard definition of frailty, its concept and potential utility is widely accepted in the MM community based on a good association with outcomes, especially in terms of mortality. Clinicians involved in the care of AL amyloidosis patients usually have the perception that most of their patients are “frail”. Our study presents and validates a new frailty scale to be used in AL amyloidosis that adequately predicts frailty-dependent mortality in this population. As far as we know, our study is the first to focus on frailty in AL amyloidosis.

We hypothesized that three frailty-associated common variables, such as age, ECOG, and NT-proBNP could be able to accurately estimate the mortality risk in AL amyloidosis patients. This triad already showed its utility as a frailty score for MM [28]. The huge relevance of NT-proBNP in AL amyloidosis has been extensively documented [35,36,37]. We adapted the cutoff of NT-proBNP to 8500 ng/L based on the current staging model [16]. The performance of this triad seems to capture the essence of frailty in these patients, both in the derivation cohort with a very high rate of heart involvement, and in the more balanced multicenter validation cohort. The behavior of the three variables is well reflected in the survival curves in both cohorts, and obviously, in the multivariate analysis, confirming their value as independent prognostic factors for mortality. With these three simple variables, every clinician involved in the care of these patients can have an easy and objective snapshot assessment of frailty from diagnosis. Moreover, all these variables change over time allowing an easy monitoring of frailty-associated risk of mortality. Thus, this core score also allows a dynamic approach, assuming that the current frailty status is a better predictor of outcomes than baseline frailty status [30]. The assessment of frailty should occupy a central position in the clinical decision-making process in the setting of systemic AL amyloidosis. 

However, this core score could be improved. Comorbidity should always be explored to assess frailty. Unfortunately, there is not a standardized approach to measure comorbidity in AL amyloidosis. A comprehensive approach could be a good option, counting all comorbidities recorded in the medical records of each patient at the time of diagnosis. We consider that CCI is not an optimal approach to measure comorbidity in this setting because this index and other scores used in MM do not take into account some relevant comorbidities.

Comorbidity added value to our Cox model, since the presence of ≥5 comorbidities in our comprehensive approach behaves as an independent prognostic factor in the multivariable analysis. Therefore, comorbidity also plays a role in the assessment of frailty in AL amyloidosis but the optimal way to include it in the frailty score remains to be determined and validated. 

Regarding early mortality, contrary to what happens in MM, in which a decrease over time is shown [38], the corresponding rate remains stable in AL amyloidosis [13]. The early mortality at 3 months was 17.2% in the derivation cohort (with 91% cardiac involvement), whereas it was 13.1% in the validation cohort, a figure almost identical to the 13.4% reported in the EMN23 observational study (2004–2018), which is the largest real-world AL amyloidosis study to date, including 4480 patients with a 68% cardiac involvement [13]. In this key study with a median follow-up similar to ours, median OS was 48.8 months (CI 95%, 45.2–51.7). Our median OS was 52.6 months (95% CI, 24.8–80.4) in the derivation cohort, despite a high rate of cardiac involvement.

The derivation cohort represents a relatively large real-world series of systemic AL amyloidosis patients managed at a Spanish referral center and included in a prospectively maintained specific registry. To the best of our knowledge, this is the largest Spanish series reported to date in the field of AL amyloidosis. As a referral center, a selection bias is probably present, confirmed by the very high rate of cardiac involvement and the high proportion of patients with advanced cardiac stages. The percentage of patients with ECOG ≥ 2 and NT-proBNP ≥ 8500 ng/L in the derivation cohort is more than twice (2.7 and 2.3, respectively) as high as in the validation cohort. Therefore, the number of frail and ultra-frail patients is correspondingly higher. Despite this, after adjusting for the frailty score, no differences can be shown in terms of OS or early mortality. This suggests that referral centers with a developed multidisciplinary care platform can have a role in the management of complex AL amyloidosis patients, ensuring a timely diagnosis, integral cardiac support, access to clinical trials, and an optimized approach to autologous stem cell transplant and heart transplant, if required. 

The prognosis assessment in AL amyloidosis is a rapidly changing field. Stage is considered the most important predictor of outcome [39]. However, the use of two different staging systems has potential for stage to be confounder [40]. Moreover, the Mayo 2012 staging system predicts late survival more accurately and the European modification predicts early mortality. Both staging systems are based on two or three well validated biomarkers, but relevant patient-associated prognostic factors are not taken into consideration, and there is a need to confirm its performance with current therapies. Therefore, a frailty assessment may complement the prognostic utility of the stage.

Our study has some limitations. The basis of the study is a retrospective observational real-world single center series. The study spans almost two decades, and comorbidity data are lacking in four patients in early years. The comorbidity is not analyzed in the validation cohort mainly due to the lack of a standardized approach to measure it.

In summary, this study highlights for the first time the importance of measuring frailty-based mortality risk in real-world systemic AL amyloidosis patients, using a simple score with three common variables. The score can be used at the time of diagnosis and during follow-up. The performance of the score was validated in an independent cohort. Comorbidity adds value to the score but the optimal strategy to analyze it remains to be determined. Overall, 38.8% in the derivation cohort and 13.1% in the validation cohort were ultra-frail patients, and the outcome of these patients was extremely poor in terms of survival and early mortality. Most patients who died before three months were ultra-frail; on the other hand, one out of five long-term survivors were also included in this category.

More and larger studies on the subject are encouraged. The field of frailty in AL amyloidosis is an opportunity to improve our knowledge of the disease and hopefully the outcome. Frail patients are commonly excluded from clinical trials. Therefore, there is an urgent need to develop specific trials for frail AL amyloidosis patients. There are no current standard recommendations for clinical decision-making in frail AL amyloidosis patients and this should be a key goal for the immediate future of clinical research in this setting.

## 5. Conclusions

The impact of frailty on mortality has been demonstrated in several hematological malignancies, particularly in MM. However, no study has so far addressed frailty and its prognostic impact in AL amyloidosis.

This study presents and validates a new frailty score for AL amyloidosis, which discriminates four groups of patients in terms of survival. The score is based on three common and frailty-trusted variables: age, ECOG and NT-proBNP. A nomogram based on the Cox model allows for easily and dynamically estimation of the risk of mortality both at diagnosis and during follow-up.

Frailty is usually applied to personalize and optimize therapy. Our study is the first attempt to measure frailty-associated mortality risk in AL amyloidosis patients, aiming to help to standardize real-life clinical-decision making.

## Figures and Tables

**Figure 1 cancers-16-01689-f001:**
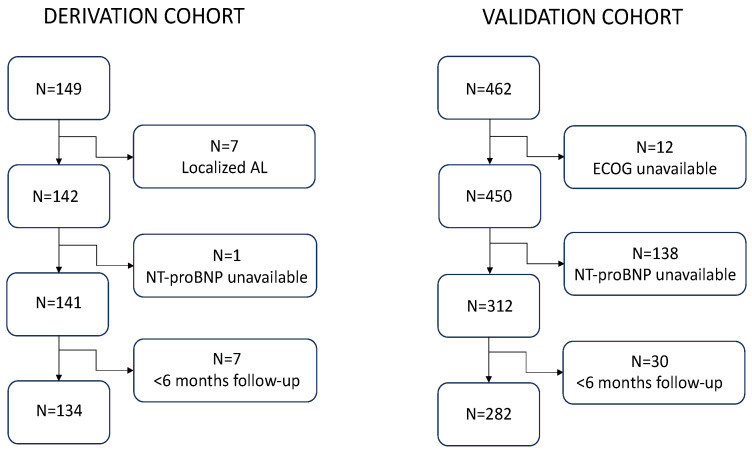
Patient selection flow chart.

**Figure 2 cancers-16-01689-f002:**
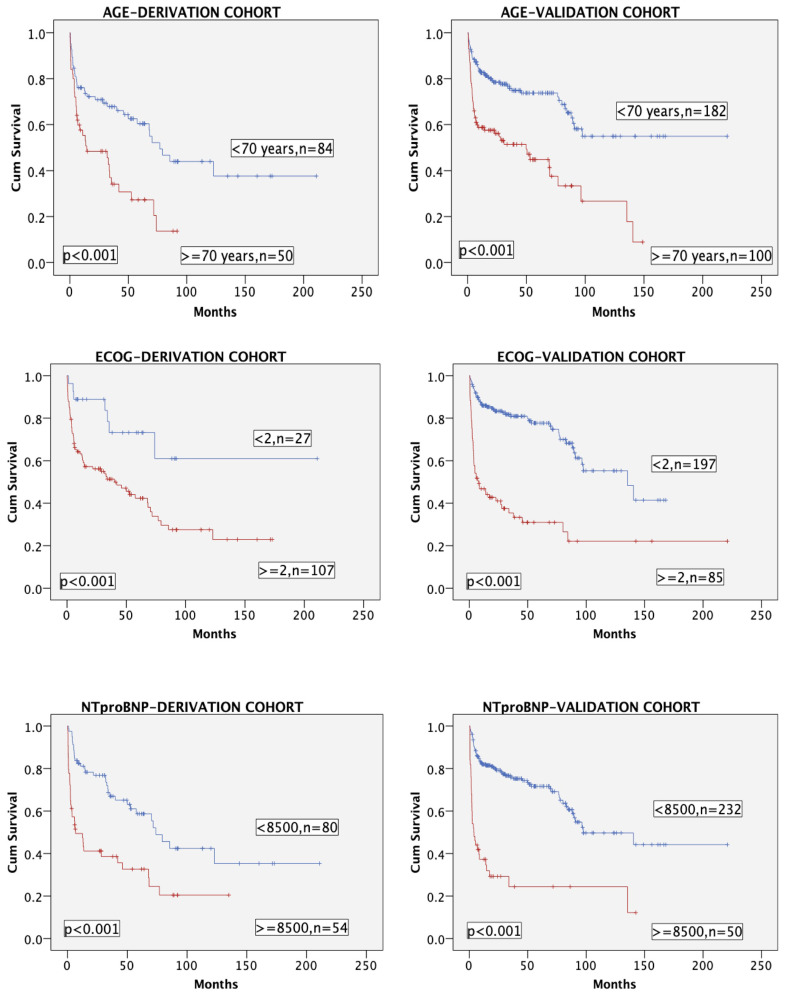
Overall survival in both cohorts according to the three variables (age < 70 vs. ≥70 years, ECOG < 2 vs. ≥2, and NTproBNP < 8500 vs. ≥8500 ng/L).

**Figure 3 cancers-16-01689-f003:**
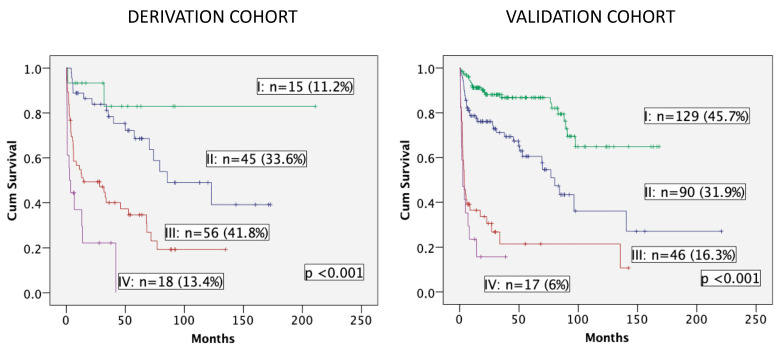
Overall survival in both cohorts according to the number of variables of the score.

**Figure 4 cancers-16-01689-f004:**
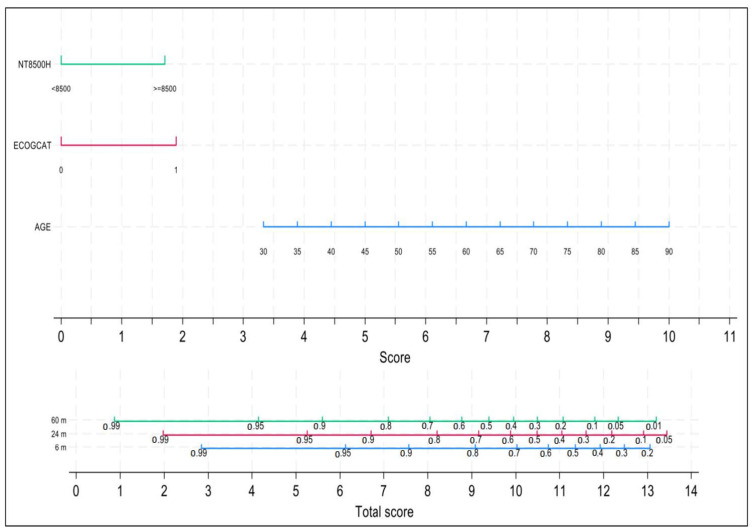
Model-based nomogram to predict survival and early mortality.

**Table 1 cancers-16-01689-t001:** Baseline characteristics of patients in the derivation cohort (n = 134).

Variables	Number of Patients (%)
Age (years) ≥ 70	50 (37.3)
Median age	64.5 (IQR 55–72.3)
Gender (male/female)	71/63 (53/47)
ECOG PS 0–1	27 (20.1)
2–4	107 (79.9)
CCI ≥ 3	76/130 (56.7)
Concurrent or previous MM	26 (19.4)
Concurrent or previous WM	4 (3)
Concurrent or previous cancer	16/129 (12.4)
BMPC (biopsy) ≥ 10%	101 (75.4)
Cytogenetic abnormalities (FISH)	12/52 (23.1)
Involved FLC, lambda	110/129 (85.3)
Heart involvement	122 (91)
Revised Staging System 2012, stage 4	56/109 (51.4)
NT-proBNP ≥8500 ng/L	54 (40.3)
≥1800 ng/L	99 (73.9)
LVEF < 45%	25/127 (19.7)
Renal involvement	63 (47)
eGFR < 60 mL/min/1.73 m^2^	52/129 (40.3)
Proteinuria ≥ 3 g/24 h	38/132 (28.8)
Renal stage 3	22/131 (16.8)
Hemoglobin < 120 g/L	36/121 (29.8)
Platelets < 120 × 10^9^/L	6/120 (5)
LDH high	56/117 (47.9)
Albumin < 35 g/L	48/122 (39.3)
Beta2-microglobulin > 5 mg/L	15/62 (24.2)
ASCT	33 (24.6)
Heart transplant	5 (3.7)

ASCT: Autologous stem cell transplant, BMPC: bone marrow plasma cells, CCI: charlson comorbidity index, ECOG PS: Eastern Cooperative Oncology Group performance status, eGFR: estimated glomerular filtration rate, FISH: fluorescence in situ hybridization [IgH + n = 6, t(4;14) n = 2, 1q + n = 2, 1p − n = 1, del17p n = 1], FLC: free light chain, LDH: lactate dehydrogenase, LVEF: left ventricular ejection fraction, MM: multiple myeloma, NT-proBNP: N-terminal pro–type-B natriuretic peptide, WM: Waldenström macroglobulinemia.

**Table 2 cancers-16-01689-t002:** Comparative characteristics of the frailty score in both cohorts.

Variables		Derivation Cohort (n = 134)	Validation Cohort (n = 282)	*p*-Value
AGE	Mean (SD)	64.4 (11.9)	64.4 (11.6)	0.970
	Median (IQR)	64.5 (55–72.2)	65 (56–73)	-
	≥70, n (%)	50 (37.3)	100 (35.5)	0.744
ECOG PS	0	0 (0)	65 (23)	<0.001
	1	27 (20.1)	132 (46.8)	<0.001
	2	57 (42.5)	54 (19.1)	<0.001
	3	44 (32.8)	29 (10.3)	<0.001
	4	6 (4.5)	2 (0.7)	<0.001
	(≥2, n (%)	107 (79.9)	85 (30.1)	<0.001
NT-proBNP	Mean (SD)	8919 (12,139)	4987.6 (7594.8)	<0.001
	Median (IQR)	6265 (1607.2–10,702.8)	1958.5 (473.8–5828.5)	-
	(≥8500 ng/L, n (%)	54 (40.3)	50 (17.7)	<0.001

ECOG PS: Eastern Cooperative Oncology Group performance status, IQR: inter-quartile range, NT-proBNP: N-terminal pro–type-B natriuretic peptide, SD: standard deviation.

**Table 3 cancers-16-01689-t003:** Univariate and multivariable Cox regression analysis.

	Univariate Analysis			Cox Regression Model		
Variables	HR	95% CI	*p*-Value	HR	95% CI	*p*-Value
Age, years	1.05	1.03–1.08	<0.001	1.06	1.03–1.08	<0.001
ECOG PS, <2 vs. ≥2	2.90	1.33–6.33	0.008	2.56	1.16–5.67	0.020
NT-proBNP, <8500 vs. ≥8500 ng/L	2.57	1.61–4.11	<0.001	2.34	1.45–3.77	<0.001
Nco, <5 vs. ≥5	2.46	1.39–4.33	0.002			
CCI, <3 vs. ≥3	1.98	1.18–3.33	0.010			
LVEF, ≥45 vs. <45%	1.99	1.14–3.48	0.016			
eGFR, ≥60 vs. <60 mL/min/1.73 m^2^	1.88	1.15–3.01	0.011			

CCI: Charlson comorbidity index, CI: confidence interval, ECOG PS: Eastern Cooperative Oncology Group performance status, eGFR: estimated glomerular filtration rate, HR: hazard rate, LVEF: left ventricular ejection fraction, Nco: number of comorbidities (comprehensive approach), NT-proBNP: N-terminal pro–type-B natriuretic peptide.

## Data Availability

All relevant data are provided within the article. The AL Amyloidosis Registry of Madrid will review requests from qualified external researchers for data in a responsible manner that includes protecting patient privacy, assurance of data security and integrity, and furthering scientific and medical innovation. Individual patient data will not be shared.

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
