# Peer review of "A Simple Frailty Score Predicts Survival and Early Mortality in Systemic AL Amyloidosis"

_cancers, 2024, doi:10.3390/cancers16091689_

Round 1

Reviewer 1 Report

Comments and Suggestions for Authors

Rios-tamayo et al have developed a simple frailty score that can predict survival and early mortality in systematic AL amyloidosis. Using 3 clinical or laboratory parameters, e.g. age, ECOG performance status, and NT-proBNP, the authors aim to develop evidence-based frailty-adapted treatment strategies for this disease. Overall, this is a very interesting study with important clinical relevance for this rare population with systematic AL amyloidosis. The study includes 134 cases as the training cohort and 282 cases as the validating cohort, which are relative big sample sizes respectively for this model. However, some routine and important laboratory parameters, such as B2 macroglobulin and cytogenetic abnormality, should be included in the analysis shown in the Table 1, this may help readers have a systematic observation about this disease. In addition, the authors may also discuss a bit why these 3 parameters show independent significance in the analysis. Is there any biological functions associate with the poor outcome for patients with the AL amyloidosis? 

Author Response

Thank you for your comments.

-Point 1: We totally agree with your suggestion to include data about beta2-microglobulin and cytogenetic abnormalities in Table 1.

As in multiple myeloma, beta2-microglobulin is a biomarker related to tumor burden with proven prognostic impact. It was available at bbaselinne only in 62 patients and this information has been added to Table 1. The inclusion in our model of the percentage of clonal bone marrow plasma cells could be also considered a variable showing the associated prognostic role of tumor burden.

Regarding cytogenetic abnormalities, the available information in the derivation cohort is scarce (only 52 patients), most patients (76.9%) having negative FISH results. This may be due on the one hand to the frequently low clonal plasma cell burden in systemic AL amyloidosis patients, and in the other hand, to methodological issues related to FISH studies, which have been mostly performed in the primary (without selection of plasma cells) bone marrow samples. This information has been added to Table 1.

-Point 2: We have emphasized in more detail in the discussion the independent prognostic role of the three frailty-associated prognostic variables of the score in the Cox model. The following sentence has been added to main text (highlighted): “The behavior of the three variables is well reflected in the survival curves in both cohorts, and obviously, in the multivariate analysis, confirming their value as independent prognostic factors for mortality”.

-Point 3: The poor outcome of systemic AL amyloidosis patients is mainly associated to the heart involvement, particularly for those patients with delayed diagnosis and advanced stages. Most of early deaths in this context are cardiovascular-related, particularly thrombosis and arrhythmias. Hence, the role of cardiologist's supporting treatment is of paramount importance, as well as reaching an early cardiac response based on the monitoring of NT-proBNP.

Reviewer 2 Report

Comments and Suggestions for Authors

Congratulations for the authors for their huge work. Amyloidosis is rare disease, but its significance is very impotant for the patient. If a doctor diagnoses with cardiac amyloidosis, especially, too late it is no way to stop progression of the disease. Because of limited costs in health care systems all over the word it is very important to choose a  patient eligable to treatment depending the stage of disease. Presented in this paper a simple frailty score which predicts survival and early mortality in systemic AL amyloidosis may be probably a simple tool to qualify patients to the optimal treatment. I think that it is important for clinical decision-making to square startimg to use a fraility score with doctors in daily practice.

Author Response

Thank you very much for your positive comments. We are convinced that a practical and dynamic assessment of frailty using this simple score in daily clinical practice will help to better define the best risk-adapted and personalized treatment for each patient in this complex scenario, trying to meet the goal of administering “the right drug at the right dose for the right patient at the right time point”. Additionally, we hope the score could also be used for specific clinical trials for frail patients.